# Effect of iron and zinc-biofortified pearl millet consumption on growth and immune competence in children aged 12–18 months in India: study protocol for a randomised controlled trial

Saurabh Mehta,[1,2,3] Julia L Finkelstein,[1,2,3] Sudha Venkatramanan,[1] Samantha L Huey,[1] Shobha A Udipi,[4] Padmini Ghugre,[5] Caleb Ruth,[6] Richard L Canfield,[1] Anura V Kurpad,[3] Ramesh D Potdar,[7] Jere D Haas[1]

For numbered affiliations see end of article.

**Correspondence to**
Dr. Saurabh Mehta;
smehta@cornell.edu

## ABSTRACT

**Introduction** Biofortified crops represent a sustainable agricultural solution for the widespread micronutrient malnutrition in India and other resource-limited settings. This study aims to investigate the effect of the consumption of foods prepared with iron- and zinc-biofortified pearl millet (FeZn-PM) by children on biomarkers of iron and zinc status, growth, and immune function.

**Methods and analysis** We will conduct a randomised controlled feeding trial in identified slums of Mumbai, India among 200 children aged between 12 and 18 months. Children will be randomised to receive foods prepared with the biofortified PM (FeZn-PM, ICTP8203-Fe) or non-biofortified PM. Anthropometric and morbidity data will be gathered every month for 9 months. Biological samples will be collected at baseline, midline and endline to assess iron and zinc status, including haemoglobin, serum ferritin, serum transferrin receptor, serum zinc, C-reactive protein and alpha-1 acid glycoprotein. Biological samples will be archived for future analyses. The midline measurement will be a random serial sample between baseline and endline. Immune function will be assessed at each time point by the measurement of T cell counts and vaccine responses in a subset, respectively.

**Ethics and dissemination** This study has obtained clearance from the Health Ministry Screening Committee of the Indian Council of Medical Research. Ethical clearance has been obtained from Cornell University's Institutional Review Board, the Inter System Biomedica Ethics Committee and St John's Research Institute's Institutional Ethics Review Board. The results of this study will be disseminated at several research conferences and as published articles in peer-reviewed journals.

**Trial registration number** Clinical trial registration number NCT02233764. CTRI registration number REF/2014/10/007731.

### Strengths and limitations of this study

► This is the first longitudinal randomised controlled trial to determine the efficacy of consuming complementary foods prepared using iron and zinc-biofortified pearl millet on both nutritional status and functional outcomes among children aged 12–18 months.

► The longitudinal random midline serial sampling strategy increases both sensitivity and the power of the proposed study, while reducing cost and invasiveness (by decreasing the number of biological samples from each participant).

► Data will be collected on iPad or laptops, using a mobile electronic data capture system framework on the iOS platform. This will decrease the potential for error in data entry compared with standard written hard copies of forms and allow direct uploading of data using a secure server.

► One limitation of this study is that complete blinding may not be possible due to potential sensory differences between the crops used in the two arms of the study.

and development of children has been demonstrated in several studies.[3–5] Similarly, existing evidence suggests that stunting is associated with zinc deficiency.[6] Suboptimal iron and zinc status also impairs immune functioning through a number of mechanisms, including a reduction in the proportion of circulating T-lymphocytes and lymphocyte proliferative responses.[7] Cognitive deficits are also observed with iron and zinc deficiency, including irreversible impairments in neurological and psychomotor development of children.[8]

### Iron and zinc biofortification

Biofortification has the potential to be a more sustainable and cost-effective approach

## INTRODUCTION
### The burden of iron and zinc deficiency

Deficiencies in iron and zinc are two of the most important public health problems globally.[1][2] The role of iron deficiency in growth

compared with other strategies such as diet diversification, fortification and supplementation to address micronutrient malnutrition among vulnerable populations.[9] In India, staple crops such as pearl millet (PM; *Pennisetum glaucum)* are consumed as part of the daily diet, particularly in Maharashtra, Gujarat, Rajasthan and Karnataka. The iron and zinc concentration in biofortified PM is reported to be 70–85 and 35–40 parts per million (ppm), respectively.[10] Previous research from southern India and preliminary data from our acceptability study indicate that the mean consumption of PM flour among Indian children is 40-60 g/day,[11 12] showing the suitability of this crop as a target for biofortification. Thus, for children, iron and zinc intakes from PM could be 8-9 mg/100 g and 4-5 mg/100 g, respectively,[10] and could help meet upto 50% of the recommended dietary allowances (RDA) for iron and zinc for children between 1 and 3 years of age. These estimates are based on the RDA recently recommended for Indians by the Indian Council of Medical Research.[13]

The primary objectives of this randomised controlled trial are to study the effect of the daily consumption of high iron and zinc-biofortified PM (FeZn-PM) on biomarkers of iron and zinc status, as defined by haemoglobin, serum ferritin, serum transferrin receptor, serum zinc, and C-reactive protein (CRP); and growth, as defined by length-for-age, weight-for-age and weight-for-length, among children aged 12–18 months, when compared with children receiving conventional PM. In addition, we will assess the effect of high FeZn-PM by 12–18 months old children on immune outcomes.

## METHODS AND ANALYSIS
### Study setting
This study will take place in identified slums of urban Mumbai, Maharashtra, India. Mumbai is the commercial capital of India, and about 41.3% of city's populace resides in urban slums.[14]

### Study design
We will conduct a randomised controlled trial in which children (12–18 months old) will be fed complementary foods prepared using FeZn-PM or the comparator conventional PM for 9 months. Caregivers will be requested to provide three informed consent forms for (1) census, (2) screening and (3) enrolment into the trial. Caregivers who provide all three consents are considered eligible to be screened. Details of the informed consent process can be found the section 'Informed consent process.'
1. Census: Before the study begins, a census will be conducted in the identified slums to gather information on the age of the child, sex and location.
2. Screening: Consent will be obtained for screening and inclusion in the trial. Non-invasive data will be collected on: dietary allergies, use of iron or zinc dietary supplements, availability of a caregiver and if the caregivers are planning to stay in the slum for the duration

of the study. Anthropometry and dietary intake will also be measured. If the children were not dewormed recently, they will be provided liquid albendazole by the study physician at the study clinic (5 mL of syrup equivalent to 200 mg per dose). If still eligible for the study, blood will be collected to measure complete blood counts including haemoglobin for further determination of eligibility.
3. Randomisation: A statistician from the Cornell Statistical Consulting Unit will generate the random allocation sequence using a statistical software package (SAS V.9.4); the randomisation sequence key will be blinded to all study personnel except the study statistician and the software developer until follow-up is over. Randomisation will be allocated at the individual level. Individuals will be randomised in blocks of 60. PM food products will coded by study arm (arm 1: FeZn-PM; arm 2: conventional PM). Children will be randomised to either study arm and will be monitored to ascertain they consume the assigned food product throughout the study duration. The midline measurement (including biological sample collection) will be a random serial sample taken at any month (months 2–7) between the baseline and endline measurements.[15] Longitudinal midline random serial sampling is often used in population pharmacokinetic research and has been shown to be a useful strategy for iron fortification efficacy studies, by describing the pattern of iron repletion.[15]

### Intervention and comparator
The intervention is FeZn-PM (ICTP-8203) developed by the International Crops Research Institute for the Semi-Arid Tropics, and the comparator is a conventional PM that is commercially available on the market. The comparator was chosen because it is similar to the intervention in most aspects, except for iron and zinc content, which allows direct analysis of the impact of the iron and zinc-biofortified variety on our outcome measurements. The nutrient composition of biofortified and the conventional PM flour of a sample lot are presented in table 1.

| Table 1 Nutrient composition of pearl millet varieties | | |
|---|---|---|
| | **Biofortified (per 100 g flour)** | **Control (per 100 g flour)** |
| Moisture (g) | 6.89 | 7.16 |
| Fat (g) | 7.88 | 8.74 |
| Protein (g) | 13.46 | 13.46 |
| Carbohydrate (g) | 30.22 | 30.98 |
| Energy (kcal) | 246 | 256 |
| Ash (g) | 1.99 | 1.63 |
| Phytate (mg) | 876.59 | 998.44 |
| Iron (mg) | 6.64 | 2.56 |
| Zinc (mg) | 4.43 | 1.24 |

We expect a child to consume an average of 40–60 g of PM per day, depending on the age of the child.[10] [11]

## Inclusion and exclusion criteria
### Inclusion criteria
Participants included in this study will be 12–18 months old male and female children with haemoglobin concentration ≥9 g/dL, living in urban slums of Mumbai.

### Exclusion criteria
Children will be excluded if: (1) they are younger than 12 months, 0 days or older than 18 months, 30 days at enrolment; (2) their haemoglobin concentration is less than 9 g/dL or haemoglobinopathy is present (as indicated via abnormal peak via haemoglobin variant analysis and confirmed by Center for the Study of Social Change (CSSC) physicians); (3) they show signs of severe malnutrition (a weight-for-length −3)[16]; (4) their caregiver reports prior known diagnosis of HIV, tuberculosis or current diagnoses of HIV, malaria, tuberculosis and/or dengue fever; (5) their caregiver is unavailable to bring the child to the feeding centre during follow-up; (6) the child has any known dietary allergies; (7) their caregivers will leave the study site for more than 4 weeks during the follow-up period and (8) prior (within the past 1 year) or current consumption of iron or zinc supplements. Children who are severely anaemic will be referred to physicians at the CSSC.

## Informed consent process
Research assistants will obtain informed consent from caregivers. Three consent forms will be collected at the screening visit: (1) prescreening consent (non-invasive questionnaires to determine age eligibility, use of dietary supplements, availability of caregiver and so on as described in exclusion criteria; (2) screening consent (invasive procedures including blood collection and anthropometric measurements to assess malnutrition); and (3) intervention consent (including baseline procedures such as a full doctor's examination and detailed background questionnaires).

## Feeding centre
Feeding centres will be located near to where children and their caregivers' dwellings are clustered, to ensure that travel time from their home to the feeding centre is within walking distance.

## Sample size estimation
Our estimates of sample size are based on assumptions about mean values and associated variation in haemoglobin and serum ferritin. We expect 50%–75% of these children to have iron and zinc deficiency based on published literature[17] and preliminary results (unpublished). We assume that absorbed iron will be transferred to body stores in the liver, and that the change in liver stores is reflected in changes in serum ferritin at a rate of 8 µg/L of serum ferritin per mg liver iron in non-anaemic, iron-depleted children. Children who have iron deficiency and were anaemic (haemoglobin <11 g/dL[18]) will have their absorbable iron directed to haemoglobin synthesis in the early months of the feeding trial. We estimate that the children consuming FeZn-PM will demonstrate a gain of 1 g/dL in haemoglobin concentrations in 2.4 months, whereas children consuming control PM will need more than 9 months to demonstrate the same increase. To detect a significant increase in ferritin from the baseline value of 1.79 (6 mg/L) with SD of 1.2 in the experimental group compared with the control group at a power of 80% and 5% significance level, 96 participants will be required.

## Follow-up
### Assessments
At baseline, we will assess anthropometry; collect blood for analysis of haemoglobin, iron, zinc, CRP, and acid glycoprotein (AGP) biomarkers in the blood (all blood analyses other than complete blood counts and T cell counts will be stored at −80 and then performed in batch at the end of the trial); nutrient intake measures using multiple-pass 24 hours dietary recall[19] for children and mothers; socioeconomic and demographic information and morbidity history. Follow-up will continue for 9 months and will include monthly anthropometric measurements, morbidity assessments, Infant and Young Child Feeding Questionnaire[20] and 24 hours dietary recall.[19] Both maternal and infant dietary data will be collected via paper forms and entered into the CS Dietary System Rel. 1.10 to calculate energy and nutrient intakes. Additionally, we will collect rectal swab or stool samples to determine microbiome composition in potential future ancillary analyses. The midline time point will be a random serial sample that occurs between the baseline and endline measures. Children may visit the CSSC clinic at any time during the trial for healthcare treatment. All blood collection procedures will include the use a local anaesthetic (Prilox-Lidocaine and Prilocaine cream) to decrease pain, and a vein finder device to illuminate the veins to better identify the injection site.

### Administration of intervention
Both arms will receive complementary foods prepared with PM three times per day, 6 days per week, for 9 months. Culturally acceptable PM-based complementary foods were developed by the Shreemati Nathibai Damodar Thackersey (SNDT) University, and the acceptability of these food products was tested on the caregivers and the children from the slums of Mumbai.[10] We will conduct a run-in/pilot phase of the study with one to three selected feeding centres.

Each day, the child's caregiver will bring the child to their feeding centre. Two meals will be consumed at the feeding centre; the third meal may be consumed at home. The food intake of the two meals at the feeding centre will be measured directly before and after consumption. To measure the intake of the third meal, the weight of unconsumed food from the third meal will be measured

the next morning at the feeding centre. To assure adherence, healthcare workers will follow-up with the participants' caregivers and record reasons for non-adherence. Throughout the study, we will conduct periodic analysis of random samples of both grain varieties and prepared food products to ensure food safety and quality of the intervention.

### Primary outcome measurements
#### Biomarkers of iron and zinc status
We will determine whether biofortified PM improves iron and zinc status compared with children who consume non-biofortified PM. Specifically, iron and zinc status will be assessed by measuring concentrations of haemoglobin, serum ferritin, serum transferrin receptor and plasma zinc at enrolment (baseline), midline and endline. In addition, we will measure concentrations of inflammatory biomarkers CRP and alpha 1-AGP, as iron and zinc biomarkers can be influenced by inflammation.

#### Growth
We will assess change in a child's growth as another outcome variable to determine whether iron-biofortified PM reduces the risk of underweight (weight-for-age z-score <−2), wasting (weight-for-height z-score <−2) and stunting (length-for-age z-score <−2) during the 9-month intervention period, compared with those who receive non-biofortified PM. We will measure weight (kg) and length (cm) at baseline and monthly throughout follow-up. The weight and length data will be converted to weight-for-age, weight-for-length z-score and length-for-age z-score. The growth rates (as kg/month and cm/month) will also be compared from the absolute measurements after controlling for age.

### Additional outcomes
#### Immune function
We will assess immune function by measuring T cell counts and vaccine responses. In addition, we will collect morbidity data, including changes in types and frequencies of morbidities such as diarrhoeal illness, pneumonia and any chronic disease throughout follow-up during each clinic visit. Caregivers will also be encouraged to come to the clinic at any time for any health-related reasons, which will also be recorded. This will provide an accurate representation of both innate and adaptive immunity in participants.

#### Cognitive function
To determine whether consumption of biofortified PM improves child cognitive function compared with consumption of non-biofortified PM, we will assess in a subset of children aspects of cognition that (1) previously have been shown to be sensitive to the effects of early iron and/or zinc deficiency; or (2) draw heavily on brain structures or processes thought to be vulnerable to early iron and/or zinc deficiency based on studies of animal models and humans. Thus, we will assess multiple specific aspects of memory, attention and processing speed using automated eye trackers.[21] In addition, we will also assess higher level, integrative cognitive abilities, including problem-solving and exploratory behaviour and global aspects of attention and inattention during free play with toys.[22]

### Data collection and storage
Prior to data collection, the field staff will be trained on ethical and data collection procedures. The protocol of data collection forms to be used is shown in figure 1. All data except dietary data will be collected on iPads or laptops for the proposed project. We will use a mobile electronic data capture (mEDC) framework on the iOS platform, ConnEDCt, specifically designed for this project.[23] A secure server will be used for uploading, storing and accessing the data. This will enable real-time feedback and error checking, as well as eliminate errors associated with data entry, facilitating faster data analysis and rapid dissemination of results. All information will be kept confidential. All names will be removed from the data for analysis. The identifying codes and linked names will be securely stored in password-protected computers. All data will be securely stored on a third-party server, which will have limited access by study team members. All biological specimens will be collected by trained medical professionals and evaluated by certified laboratories within India. Biological specimens will be stored appropriately throughout the duration of the study and after the study for future analysis.

### Data analysis
We will use an intention-to-treat approach to determine the effect of biofortified PM on the outcomes described above. Advanced analysis will use mixed models to account for PM-based complementary foods consumed and use each block of 60 as a random effect. We will also plot dose-response curves using restricted cubic splines that will help us detect any threshold effects, as well as non-linear associations. Non-parametric tests, such as the Hodges-Lehmann test, will be used where relevant, as in the case of non-normally distributed variables including serum ferritin.

### Study monitoring board
A DSMB will be established for this study and will include experts representing Cornell University and SNDT University who are not directly involved in the trial. The DSMB members will oversee the study and will periodically monitor the progress and outcomes of the intervention.

### Reporting of adverse events
All adverse events will be reported by the study physician via an adverse event/serious adverse event report form. When an adverse event occurs, study physicians, research assistants and/or other study personnel will undertake all necessary precautions to ensure the safety and well-being of the participant.

The following study protocol endpoints will be considered to define safety and efficacy outcomes and establish

**HarvestPlus Database Form Collection Protocol**

Key:
- grey = req
- crosshatch = req, daily
- vertical lines = req, random (once)
- horizontal lines = Outside EDC; manual
- dotted = not req

Requirement codes used below: R = req; D = req, daily; Ro = req, random (once); M = Outside EDC, manual; NR = not req (dotted)

| | PHASE 1 | | PHASE 2 — FOLLOW-UP | | | | | | | | | |
|---|---|---|---|---|---|---|---|---|---|---|---|---|
| **Visit #** | 1 | 2 | 3 | 4 | 5 | 6 | 7 | 8 | 9 | 10 | 11 | 12 |
| | Census | Pre-Screening, Screening | Baseline [pre-feeding] | 1 | Midline 2 | Midline 3 | Midline 4 | Midline 5 | Midline 6 | Midline 7 | 8 | 9 Endline [post-feeding] |
| **Phase 1 – Screening** | | | | | | | | | | | | |
| Census (OED) | M | | | | | | | | | | | |
| *Consent for Prescreening* | | R | | | | | | | | | | |
| Prescreening/RA Form | | R | | | | | | | | | | |
| *Consent for Screening* | | R | | | | | | | | | | |
| Screening_Lab/Clinic | | R | | | | | | | | | | |
| Screening_Anthropometry | | R | | | | | | | | | | |
| Screening_RectalSwab | | R | | | | | | | | | | |
| Screening_WHZ_Zscore | | R | | | | | | | | | | |
| *Consent for Trial* | | R | | | | | | | | | | |
| **Phase 2 – Trial** | | | | | | | | | | | | |
| FollowUp_Baseline_SES | | | R | | | | | | | | | |
| FollowUp_Lab | | | | | Ro | Ro | Ro | Ro | Ro | Ro | | R |
| FollowUp_RectalSwab | | | | | Ro | Ro | Ro | Ro | Ro | Ro | | R |
| FollowUp_Cognition | | | | | Ro | Ro | Ro | Ro | Ro | Ro | | |
| FollowUp_Morbidity | | | R | R | R | R | R | R | R | R | | R |
| FollowUp_Anthropometry | | | R | R | R | R | R | R | R | R | | R |
| FollowUp_Feeding1 | | | | R | R | R | R | R | R | R | | |
| FollowUp_TB_Contact_Investigation | | | R | R | R | R | R | R | R | R | | R |
| FollowUp_IYCF | | | R | R | R | R | R | R | R | R | | R |
| FollowUp_FFQ | | | M | M | M | M | M | M | M | M | | M |
| FollowUp_Recall | | | M | M | M | M | M | M | M | M | | M |
| AER {Adverse Event Report} | | | NR | NR | NR | NR | NR | NR | NR | NR | NR | NR |
| DropOut {Termination} | | | NR | NR | NR | NR | NR | NR | NR | NR | NR | NR |
| Other Issue {Not in draft form database} | | | NR | NR | NR | NR | NR | NR | NR | NR | NR | NR |

**Notes:**
The midline measurements will take place in months 2–7.

**Figure 1** Form collection protocol. AER, adverse event report; EDC, electronic data capture; FFQ, food frequency questionnaire; IYCF, infant and young child feeding; OED, outside electronic database; RA, research assistant; SES, socioeconomic status; TB, tuberculosis; WHZ, weight-for-height Z-score.

unblinding and stopping guidelines in this trial, as per the discretion of the DSMB: (1) diagnosis of the development of severe acute malnutrition such as Kwashiorkor; (2) occurrence of all-cause death. Demonstration of efficacy, namely a significant beneficial effect on mortality and other adverse outcomes will be used as a guideline to determine w the study should be unblinded, stopped or terminated.

### Ethics and dissemination

Before conducting interviews or allowing participation in the study, written informed consent will be obtained from each participant's guardians/primary caregivers and research assistants will record each granting of informed consent using audio and video technology. Severely anaemic (<7 g/dL) and malnourished children will be provided with appropriate medical care and referred to CSSC. The caregivers will be assured of the confidentiality and anonymity of reports and publications generated from this study. Participation will be voluntary, and participants will be assured that refusal to participate in the study will not impact their access to care. This study has received clearance from Health Ministry's Screening Committee of the Indian Council of Medical Research (ICMR). The results of this study will be disseminated at several research conferences and as published articles in peer-reviewed journals. The present study protocol was prepared in accordance to the Standardized Protocol Items: Recommendations for Intervention

Trials (SPIRIT) statement.[24] This trial has been registered at Clinicaltrials.gov (registration number NCT02233764) and the Clinical Trials Registry of India (CTRI) (reference number REF/2014/10/007731, CTRI number CTRI/2015/11/006376).

**Author affiliations**
[1]Division of Nutritional Sciences, Cornell University, Ithaca, New York, USA
[2]Institute for Nutritional Sciences, Global Health, and Technology, Cornell University, Ithaca, New York, USA
[3]Division of Nutrition, St John's Research Institute, Bangalore, Maharashtra, India
[4]Kasturba Health Society Medical Research Centre (KHS-MRC), Mumbai, Maharashtra, India
[5]Department of Food Science and Nutrition, Shreemati Nathibai Damodar Thackersey, Women's University (SNDT), Mumbai, India
[6]Data Performance LLC, New York, USA
[7]Center for the Study of Social Change, Mumbai, India

**Contributors** SM, JLF and JDH designed the study, conceived the research questions and prepared the study protocol with feedback from the study collaborators. SM is the principal investigator of the study and JLF, JDH, SAU, PG, RLC, AVK and RDP are coinvestigators. CR developed the database used in this trial and provided feedback on the protocol. SLH and SV contributed to the editing of the protocol and will supervise the data collection under the guidance of the investigators. All authors read and approved the final protocol.

**Funding** This work was supported by HarvestPlus, grant number 2014H8302.

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
