## [Reviewer comments · BMJ Open]

ARTICLE DETAILS

TITLE (PROVISIONAL)	Effect of Iron- and Zinc-Biofortified Pearl Millet Consumption on Growth and Immune Competence in Children Aged 12-18 Months in India—Study Protocol for a Randomised Controlled Trial
AUTHORS	Mehta, Saurabh; Finkelstein, Julia; Venkatramanan, Sudha; Huey, Samantha; Udipi, Shobha; Ghugre, Padmini; Ruth, Caleb; Canfield, Richard; Kurpad, Anura; Potdar, Ramesh; Haas, Jere

VERSION 1 – REVIEW

REVIEWER	Yunus Emre Tuncil Whistler Center for Carbohydrate Center Food Science Department Purdue University West Lafayette, IN 47907 USA
REVIEW RETURNED	10-Jun-2017

GENERAL COMMENTS	The manuscript has been written clearly and describes well the study objective. I only have few comments as indicated below point by point; 1) In the Methods and analysis section of the abstract (page 2, line 43), please be specific rather than saying other biomarkers.2) In page 5, line 101; it is not clear whether 61-80 g/day is average consumption of PM by Indian children or it is a global average. Please indicate this3) In page 5 line 102, reference is needed at the end of the sentence (after respectively)4) In page 5 line 104, please include the recommended RDA for PM consumption by children along with its reference.5) Generally, the introduction has been written nicely. It will be actually good to give more explanation about the biofortified PM (or other grains) in the introduction. For example, how often are they consumed? What is the differences between conventional grains and biofortified grains (i.e. is there any chemical differences in terms of protein, carbohydrate, fiber, etc content) which could potentially affect their health effects.6) Page 7 lines 140 - 142; how will the biological samples collected for the screening be stored until the time of analysis (-80, or -50) or will they be immediately analyzed? It is also needed to include this for the samples collected during the study.
---

	7) Page 7; Please clearly indicate whether children who fails in the first screening will be screened in the "screening phase 2" or they will be excluded 8) Page 7, line 145; what is the statistical software package that you propose to use. Please be specific. 9) Page 11, line 251; what are the complementary foods? Do you have any criteria for selecting the complementary foods? Have you had a chance to search whether zinc and iron could potentially react with any food components, which could decrease/increase their efficiency on health outcomes? I would suggest you to choose complementary foods accordingly, and please include your selection criteria for complementary foods. 10) During the biofortification process, the composition of PM (protein, carbohydrate, fiber, vitamin, minor nutrients, etc) can be affected. Therefore, it will be actually perfect to chemically analyze the conventional PM and biofortified PM used in this study in terms of protein content, carbohydrate content, fiber content, and minor nutrient content. These are relatively easy analysis, but, I believe, will dramatically increase the value of the study. 11) It will be good to discuss more about the expected results.
--	--

REVIEWER	Nicola Lowe University of Central Lancashire, UK
REVIEW RETURNED	22-Jul-2017
GENERAL COMMENTS	Clear and concise. No revisions required.

VERSION 1 – AUTHOR RESPONSE

Reviewer 1

Author Response - Thank you for your detailed review of the manuscript - we appreciate the time and your feedback!

Comment 1) In the Methods and analysis section of the abstract (page 2, line 43), please be specific rather than saying other biomarkers.

Response: Please see lines 42-43.

Comment 2) In page 5, line 101; it is not clear whether 61-80 g/day is average consumption of PM by Indian children or it is a global average.

Response: Please indicate this. Please see line 98-101.

Comment 3) In page 5 line 102, reference is needed at the end of the sentence (after respectively)

Response: Please see the new reference at the end (after “respectively”).

Comment 4) In page 5 line 104, please include the recommended RDA for PM consumption by children along with its reference.

Response: Information on recommended dietary allowances is included in lines 103-106.

Comment 5) Generally, the introduction has been written nicely. It will be actually good to give more explanation about the biofortified PM (or other grains) in the introduction. For example, how often are they consumed? What is the differences between conventional grains and biofortified grains (i.e. is there any chemical differences in terms of protein, carbohydrate, fiber, etc content) which could potentially affect their health effects.

Response: Please see the revised Introduction section; revisions are highlighted in yellow. Please also see the addition of a new Table 1 (line 170). An acceptability study was conducted (recently published) to select the foods for inclusion in the study. This manuscript includes more details on the selection of foods and their preparation.

Comment 6) Page 7 lines 140 - 142; how will the biological samples collected for the screening be stored until the time of analysis (-80, or -50) or will they be immediately analyzed? It is also needed to include this for the samples collected during the study.

Response: Please see lines 141-142 and line 234

Comment 7) Page 7; Please clearly indicate whether children who fails in the first screening will be screened in the "screening phase 2" or they will be excluded

Response: Information on screening and randomization is included in lines 132-144, and inclusion and exclusion criteria is included in lines 179-196

Comment 8) Page 7, line 145; what is the statistical software package that you propose to use. Please be specific.

Response: Information on statistical package is included in line 145

Comment 9) Page 11, line 251; what are the complementary foods? Do you have any criteria for selecting the complementary foods? Have you had a chance to search whether zinc and iron could potentially react with any food components, which could decrease/increase their efficiency on health outcomes? I would suggest you to choose complementary foods accordingly, and please include your selection criteria for complementary foods.

Response: Details on complementary foods preparation is included in lines 251-256; foods were chosen based on previous studies, including the recently published acceptability study (<http://journal.frontiersin.org/article/10.3389/fnut.2017.00039/full>), which are referenced.

Comment 10) During the biofortification process, the composition of PM (protein, carbohydrate, fiber, vitamin, minor nutrients, etc) can be affected. Therefore, it will be actually perfect to chemically analyze the conventional PM and biofortified PM used in this study in terms of protein content, carbohydrate content, fiber content, and minor nutrient content. These are relatively easy analysis, but, I believe, will dramatically increase the value of the study.

Response: Chemical composition of both the varieties were similar except for iron and zinc concentrations. Nutrient composition of pearl millet are now presented in Table 1 (line 170) and lines 163-165.

Comment 11) It will be good to discuss more about the expected results.

Response: Please see previous studies we have referenced.

VERSION 2 – REVIEW

REVIEWER	Yunus Emre Tuncil Whistler Center for Carbohydrate Research, Food Science Department, Purdue University, West Lafayette, IN 47907 USA
REVIEW RETURNED	07-Sep-2017
GENERAL COMMENTS	The authors have appropriately addressed the suggestions, which, I think, resulted in a further improvement of the manuscript.